# Post-Vaccination *Streptococcus pneumoniae* Carriage and Virulence Gene Distribution among Children Less Than Five Years of Age, Cape Coast, Ghana

**DOI:** 10.3390/microorganisms8121987

**Published:** 2020-12-13

**Authors:** Richael O. Mills, Mohammed R. Abdullah, Samuel A. Akwetey, Dorcas C. Sappor, Isaac Cole, Michael Baffuor-Asare, Johan A. Bolivar, Gustavo Gámez, Mark P. G. van der Linden, Sven Hammerschmidt

**Affiliations:** 1Department of Molecular Genetics and Infection Biology, Interfaculty Institute for Genetics and Functional Genomics, Center for Functional Genomics of Microbes, University of Greifswald, 17487 Greifswald, Germany; richael.mills@ucc.edu.gh (R.O.M.); Mohammed.Abdullah@med.uni-greifswald.de (M.R.A.); 2Department of Biomedical sciences, University of Cape Coast, Cape Coast, Ghana; samuel.akwetey@stu.ucc.edu.gh (S.A.A.); dorcaschristiana20@gmail.com (D.C.S.); 3Department of Microbiology, Central Laboratory Service, Korle Bu Teaching Hospital, Accra 00233, Ghana; Iykecole@gmail.com (I.C.); mikebaffuor@rocketmail.com (M.B.-A.); 4Basic and Applied Microbiology (MICROBA) School of Microbiology, University of Antioquia Medellin, 050010 Medellin, Colombia; alexis.bolivar@udea.edu.co (J.A.B.); gustavo.gamez@udea.edu.co (G.G.); 5German National Reference Center for Streptococci, Department of Medical Microbiology, University Hospital RWTH Aachen, 52074 Aachen, Germany; mlinden@ukaachen.de

**Keywords:** pneumococcal carriage, serotypes, PCV13, antibiotic resistance, virulence genes, Ghana

## Abstract

In 2012, Ghana introduced PCV13 into its childhood immunization program. To monitor the pneumococcus after PCV13 vaccination, we analyzed serotypes, antibiotic resistance, and virulence genes of pneumococcal carriage isolates among children under five years of age. We obtained nasopharyngeal swabs from 513 children from kindergartens and immunization centers in Cape Coast, Ghana. Pneumococcal serotypes were determined by multiplex-PCR and Quellung reaction. Antibiotic resistance and virulence genes prevalence were determined by disc diffusion and PCR respectively. Overall, carriage prevalence was 29.4% and PCV13 coverage was 38.4%. Over 60% of the isolates were non-PCV13 serotypes and serotype 23B was the most prevalent. One isolate showed full resistance to penicillin, while 35% showed intermediate resistance. Resistance to erythromycin and clindamycin remained low, while susceptibility to ceftriaxone, levofloxacin and vancomycin remained high. Penicillin resistance was associated with PCV13 serotypes. Forty-three (28.5%) strains were multidrug-resistant. Virulence genes *pavB*, *pcpA*, *psrP*, *pilus-1*, and *pilus-2* were detected in 100%, 87%, 62.9%, 11.9%, and 6.6% of the strains, respectively. The pilus islets were associated with PCV13 and multidrug-resistant serotypes. PCV13 vaccination had impacted on pneumococcal carriage with a significant increase in non-PCV13 serotypes and lower penicillin resistance. Including PcpA and PsrP in pneumococcal protein-based vaccines could be beneficial to Ghanaian children.

## 1. Introduction

The highest burden of infections caused by *Streptococcus pneumoniae* can be found in low and middle-income countries (LMIC). About 50% of the >250,000 global pneumococcal deaths were reported from LMIC [1]. *S. pneumoniae*, also known as the pneumococcus, is an important pathogen associated with severe and mild infections in children and the elderly. A huge burden of pneumococcal disease is registered in children, especially those under five years of age. Common non-invasive pneumococcal infections that occur in children include otitis media, sinusitis, and pneumonia. However, children can also suffer from invasive pneumococcal diseases (IPD), such as septicemia and meningitis [1,2]. Children are known to be reservoirs of the pneumococcus as they tend to carry the bacterium in their nasopharynx [3]. Pneumococcal colonization precedes pneumococcal disease. Therefore, carriage isolates can be used to predict characteristics of pneumococcal isolates in IPD. Although invasive disease surveillance systems are recommended for monitoring the pneumococcus, in LMIC, where such systems are weak or non-existent, pneumococcal carriage studies could serve as an alternative and uncomplicated means of studying the pneumococcus and the effect of vaccination [4,5]. PCV7 was the first pneumococcal conjugate vaccine that was introduced in 2000 but was never introduced in Ghana. Subsequently, PCV10 and PCV13 were introduced in 2009. The latter is currently in use in 129 countries [1]. The success of PCV vaccination was described by the drastic annual decline of 8% in pneumococci-related deaths recorded between 2010 and 2015 (Wahl et al., 2018). As a result, PCV vaccination is expected to further reduce pneumococcal carriage and disease among Ghanaian children as shown in other studies [6,7]. Other benefits of PCV vaccination are embedded in its ability to reduce antibiotic-resistant strains as well as confer herd protection [7,8]. In May 2012, Ghana introduced PCV13 vaccination into its childhood immunization program based on a vaccine schedule of 3 + 0 [9]. In 2019, Ghana reported PCV13 vaccination uptake of 97% [9]. Two pre-vaccination carriage studies, conducted one year before the implementation of PCV13 in Ghana, reported carriage rates of 34% and 49% in children who are younger than six years [10,11]. Serotypes 19F, 23F, 14, 6A, and 6B detected in the two pre-vaccination studies showed multidrug resistance. The benefits of PCV vaccination have been counteracted by serotype replacement, and serotype-dependent protection has been widely reported in both vaccinated and unvaccinated children [8,12,13]. The success of PCVs can be supplemented by potential protein-based vaccines that can offer broader protection across several serotypes. Indeed, many of these protein candidates are in different stages of clinical trials [14]. Such proteins should be highly conserved in all pneumococci, show minimal genetic diversity, be immunogenic, and safe. Promising candidates include pneumococcal choline-binding protein A (PcpA), pneumolysin (Ply), autolysin A (LytA), pneumococcal serine-rich repeat protein (PsrP), the pili, the poly-histidine triad (Pht) proteins, and lipoproteins *SP_2180*, *SP_0148* [14,15,16]. PcpA promotes attachment to epithelial cells and its presence is necessary for lung infection. Similar to PcpA, the PsrP protein is involved in pneumococcal aggregation and adhesion to lung epithelial cells [17]. PsrP is the largest surface protein in pneumococci and is required for full virulence. Rose and colleagues showed that anti-PsrP antibodies are protective in mice [15]. Two types of pili have been identified in pneumococci namely Pilus islets 1 and 2. These surface-exposed hair-like structures promote nasopharyngeal colonization and activate complement [18]. Other studies have illustrated the ability of PCV vaccination to reduce piliated strains in a pneumococcal population [19]. Despite the progress made in detecting the efficacy of these protein candidates, very few data exist on their prevalence in pneumococcal carriage isolates. However, the availability of such data would provide valuable information for protein-based vaccine formulations in the future.

Six years after the implementation of PCV13 in Ghana, limited data are available to describe the impact of PCV13 vaccination among Ghanaian children [7,20]. One study that investigated the impact of PCV13 among healthy children was conducted among children mainly residing in Accra, which is the capital city of Ghana. However, children below 24 months of age were under-represented and the prevalence of virulence genes was not determined [20]. Therefore, this study aims to determine the prevalence of virulence genes among carriage strains. It also estimates pneumococcal carriage among children in the Cape Coast metropolis of the Central region of Ghana, a setting where no pre- or post-vaccination data existed prior to this study. Furthermore, we assessed the impact of PCV13 on serotype distribution and antibiotic susceptibility among the pneumococcal isolates.

## 2. Materials and Methods

### 2.1. Description of Study the Site

This study was conducted in Cape Coast located in the Central region of Ghana. Cape Coast is a coastal town that is bounded on the south by the Gulf of Guinea, with a characteristic humid and warm climate. It is the smallest district in Ghana with a population of nearly 170,000 with children 0–9 years forming 19.1% of the population [21]. Like in other parts of Ghana, children in Cape Coast are eligible for PCV13 vaccination, which has been incorporated into the childhood immunization program. The current PCV13 coverage in Ghana is estimated at 97% following a dosing schedule of 6, 10, and 14 weeks, respectively [9].

### 2.2. Study Participants

This cross-sectional study was carried out in February 2018. Children attending kindergartens and immunization clinics in Cape Coast were invited to participate. First, a list of all kindergartens was obtained from the Central regional office of the Ghana Education Service (GES). The GES granted a letter to the research team, which was presented to all heads of kindergartens. Second, written consent was obtained from parents who agreed to allow their children to participate in the study. The majority of parents also participated in informational meetings organized by researchers in conjunction with heads of kindergartens and immunization centers. At these meetings, the objectives, benefits, and the overall concept of the study were presented to parents prior to giving their consent. Only children whose parents gave their consent are included in the study. However, if a child declined to participate or showed visible signs of respiratory infection such as runny nose or cough at the time of sample collection, this child was also excluded from the study. Basic demographic data and vaccination status were collected from each child on the day of sample collection. In total, 6 kindergartens and 6 immunization clinics were involved in this study.

### 2.3. Sample Collection and Transport

Copan flexible minitip flocked swabs (FLOQSwab™) were used to obtain samples from the nasopharynx of the children. All nasopharyngeal (NP) samples were collected by trained health care personnel. The procedure for sample collection was based on the method described by the World Health Organization (WHO) Pneumococcal working group [22]. Briefly, the parents or guardian were guided to securely hold the child’s head such that it was slightly tilted backward. The swab was introduced into the nasopharynx about one-half to two-thirds the distance from the nostril to the ear lobe. It was gently rotated at 180° or left in place for 5 s then removed slowly and aseptically placed in 1 mL of sterilized skimmed milk–tryptone–glucose–glycerin (STGG) transport and storage medium [22]. The samples were transported on ice to the laboratory where they were vortexed for 5–10 s and stored in a −80 °C freezer for further processing.

### 2.4. Pneumococcal Isolation

Pneumococcal identification was based on the WHO recommended guidelines [22]. In brief, frozen NP samples were thawed, vortexed for 10–20 s, and 10 μL of every sample was inoculated on 5% defibrinated sheep blood agar supplemented with 5 μg/mL gentamicin (Roth). The inoculated agar plates were incubated overnight at 37 °C in 5% CO_2_. α-hemolytic colonies were selected and tested for optochin susceptibility. An isolate was identified as *Streptococcus pneumoniae* (SP) when it showed susceptibility to optochin (diameter of inhibition zone ≥14 mm). However, isolates with a zone of inhibition between 7–13 mm were subjected to a bile solubility test. Bile soluble strains and those susceptible to optochin were transported in Amies transport medium (Sarstedt, Nümbrecht, Germany) to the Center for Functional Genomics of Microbes (C_FunGene, Greifswald, Germany), Greifswald, Germany for further analysis. At C_FunGene, the swabs from the transport medium were inoculated again on 5% sheep blood agar (Oxoid, Germany) and incubated overnight at 37 °C in 5% CO_2_. The isolates were again verified as pneumococci using standard microbiological techniques. A single colony was selected from each agar plate and stored in STGG at -80 °C.

### 2.5. Antibiotic Susceptibility Testing

Antibiotics discs (Thermo Fisher, Basingstoke, UK) containing 1 μg oxacillin (OX), 30 μg chloramphenicol (CHL), 15 μg erythromycin (ERY), 25 μg cotrimoxazole (COT), 30 μg tetracycline (TET), 2 μg clindamycin (DA), 30 μg linezolid (LZD), 30 μg ceftriaxone (CRO), 30 μg vancomycin (VA), and 5 μg levofloxacin (LEV) were used. Briefly, pneumococcal suspension equivalent to 0.5 McFarland standard was inoculated on a Müller-Hinton blood agar supplemented with 5% sheep blood (Oxoid, Germany). Antibiotic discs were applied on the agar plates followed by incubation at 37 °C in 5% CO_2_ for 18–24 h. *S. pneumoniae* ATCC 49619 was included in each test batch as a control strain. Antibiotic susceptibility was determined based on the CLSI interpretative chart [23]. Isolates with inhibition zone diameter of ≤19 mm to 1 μg oxacillin were considered non-susceptible to penicillin and therefore penicillin G E-test strips (Liofilchem, Waltham, MA, USA) were used to screen the strains for penicillin susceptibility. Isolates with MIC ≤0.06 μg/mL, 0.12–1 μg/mL, and ≥2 μg/mL were defined as penicillin-susceptible, penicillin-non-susceptible and penicillin-resistant respectively. Isolates that showed resistance to ≥3 classes of antibiotics were classified as multidrug-resistant (MDR).

### 2.6. Characterization of S. pneumoniae Isolates

Genomic DNA was extracted from pneumococcal cells using QIAamp DNA Mini Kit (Qiagen, Hilden, Germany). The extracted DNA for each isolate was used as DNA template for all molecular tests.

#### 2.6.1. Serotyping

Multiplex PCR (mPCR) and Quellung reaction were the two methods used to determine *S. pneumoniae* serotypes. The PCR protocol and primer pairs described by the CDC for deducing pneumococcal serotypes for African samples were used [24]. To distinguish between members of serogroup 6, we used primer pairs described previously [25]. Eight reactions were set up for each isolate. Each reaction pool contained five carefully selected primers representing different serotypes to yield stepwise PCR fragment sizes. In addition, primers for the capsular polysaccharide synthesis gene A (*cpsA*) were included in each reaction as a positive control. Each reaction mixture contained 1 μL DNA extract, 1 μL of 25 mM MgCl_2_ (Roth), 1 μL of 5 mM dNTPS (Thermofisher, Carlsbad, California), 2.5 μL of 10× Dream buffer (Thermofisher), 0.5 μL of DreamTaq DNA polymerase (Thermofisher), 1 μL of *cpsA* gene-specific primers, varying quantities of forward and reverse primers for each of the serotypes, and nuclease-free water to make a final volume of 25 μL [24]. The PCR reactions were run in a T3 Thermal Cycler (Biometra, Göttingen, Germany) and programmed as follows: 4 min at 94 °C followed by 30 cycles composed of 30 s at 94 °C, 30 s at 55 °C, 60 s at 72 °C and 5 min at 72 °C. Serotype deduction was performed by visualizing mPCR products on 2% agarose gel. Further, serotype-specific primers were used in single PCR reactions to confirm serotypes deduced from the mPCR. The Quellung reaction was applied to strains that could not be serotyped and those that could only be detected to the serogroup level by mPCR. This procedure was carried out at the German National Reference Center for Streptococci, Aachen Germany.

#### 2.6.2. Determination of Virulence Genes

Each isolate was screened for the presence of virulence genes such as *lytA*, *pavB*, *pcpA*, *psrP*, pilus islets (PI) *PI-1*, and *PI-2* by PCR using primer pairs described in this study (Appendix A). Primers targeting *rrgA* (3’-end) and *rrgB* (5’-end) amplified *PI-1*, while those targeting *sipA* (3’-end) and *pitB* (5’-end) amplified *PI-2*. Primers for *psrP* targeted the non-repetitive basic region. To confirm the absence of these genes, we used primers that targeted the flanking regions. Each reaction mixture contained 1 μL DNA extract, 1 μL of 25 mM MgCl2 (Roth, Karlsruhe, Germany), 1 μL of 5 mM dNTPS (Thermofisher), 2.5 μL of 10× Dream buffer (Thermofisher), 1 μL of the respective primers, 0.5 μL of DreamTaq DNA polymerase (Thermofisher), and nuclease-free water to make up 25 μL end volume. The peqSTAR 2× thermocycler (VWR) was used under these conditions: 4 min at 94 °C followed by 30 cycles composed of 30 s at 94 °C, 30 s at 55 °C, 1–3 min at 72 °C (varies based on expected amplicon size) and 5 min at 72 °C. A 0.8% agarose gel was used to visualize the PCR fragments.

### 2.7. Statistical Analysis

Data were analyzed with SPSS version 21 (IBM company, Armonk, New York; USA) and Graph Pad Prism version 5 (Graph Pad software). Descriptive statistics were performed on demographic data. Categorical variables were expressed as proportions and compared using the Chi-square test or Fisher’s exact test (two-tailed). The confidence interval (CI) was set at 95% for univariate and multivariate analysis and a *p*-value of <0.05 was considered statistically significant. Logistic regression analysis was used to estimate crude and adjusted odds ratios.

### 2.8. Ethical Approval

The study was approved by the institutional review board of the University of Cape Coast (UCCIRB/EXT/2017/21). Parents and guardians signed an informed consent form before a child was included in the study. Permission to conduct the study was obtained from the Ghana Education Service, Cape Coast.

## 3. Results

### 3.1. Characteristics of the Study Participants

With a nearly balanced gender distribution, males made up 51.1% (*n* = 262) and females 48.9% (251) of the total 513 participating children. The median age of the group was 30 months with a range of 6 to 59 months. Four hundred and thirteen (80.5%) of the children were selected from kindergartens while 100 (19.5%) were from immunization centers. All the children were fully vaccinated with PCV13. Participant characteristics are summarized in Table 1.

### 3.2. Pneumococcal Carriage in Children

*S. pneumoniae* was found in 151 carriage isolates bringing the overall pneumococcal carriage prevalence to 29.4%, (95% CI 0.25–0.33). The carriage prevalence among children from kindergartens and immunization centers was nearly equal with no difference in carriage prevalence by gender (Table 1). Although there was no significant difference between carriage and various age groupings, the highest carriage prevalence (38%, 95% CI 23–53) was observed in age group 43–48 months.

### 3.3. Pneumococcal Serotypes and PCV13 Serotypes Identified in Carriage Isolates

Twenty-five different serotypes were detected in addition to one non-typeable strain. The predominant serotypes observed were serotypes 23B (*n* = 22, 14.6%), 6B (*n* = 14, 9.3%), 23F (*n* = 13, 8.6%), 13 (*n* = 11, 7.3%) and 19F (*n* = 11, 7.3%) (Figure 1). Children within age groups 25–30 months, 31–36 months, and 43–48 months each had 12 different serotypes. Overall, PCV13 serotype carriage prevalence was 11.3% (58/513) and PCV13 coverage was of 38.4% (58/151). However, only 30.5% of the serotypes were covered by PCV10 while PPV23 coverage was 58.9%. Although PCV13 serotype carriage was not significantly associated with the different age groups, children within age groups 25–45 months mostly carried PCV13 serotypes. On the contrary, non-PCV13 serotypes covered 61.6% (*n* = 93) of the isolates and they were detected among all age groups except in 6 months old children.

### 3.4. Antibiotic Susceptibility of Carriage Isolates

All 151 isolates were fully susceptible to linezolid, vancomycin, and levofloxacin. Low non-susceptibility prevalence was observed against ceftriaxone, clindamycin, erythromycin, and chloramphenicol 2.6%, 5.3%, 7.3%, and 11.3%, respectively. In contrast, resistance to tetracycline and cotrimoxazole was above 55%. Only one isolate was fully resistant to penicillin while 35% of the strains showed intermediate resistance (Figure 2). Penicillin resistance was significantly associated with PCV13 serotypes. While the majority of non-PCV13 serotypes were susceptible to penicillin, penicillin resistance was observed among non-PCV13 serotypes 23B, 15A, 35B, and 38 (Figure 3). Forty-three (28.5%) of the isolates showed intermediate or full resistance to three different classes of antibiotics and considered to be multidrug-resistant (Figure 4), while 23 (15.2%) were resistant to ≥4 classes of antibiotics. However, the serotype distribution within the ≥3 and ≥4 MDR groups remained the same. Additionally, there was a significant association between multidrug resistance and PCV13 serotypes (OR 4.19, 95% CI 1.99–8.84) (Table 2).

### 3.5. Virulence Genes Detected in Carriage Isolates

The *cpsA, lytA*, and *pavB* virulence genes were detected in all 151 pneumococcal isolates. However, variations were observed in the distribution of the other virulence genes (Figure 5). The prevalence of *pcpA*, *psrP*, *PI-1*, and *PI-2* were 132 (87%), 95 (62.9%), 18 (11.9%), and 10 (6.6%), respectively.

Whereas *pcpA* and *psrP* were well represented among both PCV13 and non-PCV13 serotypes, the pilus islets were disproportionately present among PCV13 serotypes (Table 3). *PI-1* was identified in serotypes 6A, 6B, 19F, 9V, 13, and 11A while *PI-2* was only present in serotypes 23B and 19F. Five serotype 19F strains possessed both *PI-1* and *PI-2*. *PI-1* and *PI-2* were associated with antibiotic resistance (OR 4.22, *p* = 0.032, OR 2.24, *p* = 0.16), while *pcpA* and *psrP* were not (Table 2).

## 4. Discussion

This is the first pneumococcal carriage study to be carried out in Cape Coast, Ghana. In this study, the pneumococcal carriage prevalence was 29.4%, which is lower when compared to the 54% [20] obtained from children attending nurseries and kindergartens in Accra, Ghana. This difference in carriage prevalence could be attributed to the variation in age distribution as well as the settings from which these children were recruited. Despite these differences the carriage prevalence obtained in this study corroborates with other post PCV studies [26,27]. Furthermore, an early post PCV13 study performed in Tanzania, a country that also introduced PCV13 in the same year as Ghana, reported a carriage prevalence of 31% among children ≤24 months [28], similar to the 30.9% reported from this study among children in the same age group. Contrastingly, the drastic reduction in carriage reported by high-income countries and some middle-income countries following PCV vaccination was not observed in this study [29,30]. Particularly, a more recent study from Turkey, a country that switched from PCV7 to PCV13 in 2011, reports of very low PCV carriage prevalence of 11.8% and 8.4% among children <24 months and 24–59 months, respectively [29]. However, higher post PCV carriage have been seen in Nigeria, Malawi, Belgium, Norway, Russia, and Laos [6,13,31,32,33]. Two pre-vaccination studies in Ghana reported carriage prevalence of 34% and 49% [10,11]. Comparing these pre-vaccination data with post-vaccination carriage prevalence of 29.1% suggests that PCV vaccination in Ghana has not had a greater impact on pneumococcal carriage. This phenomenon where no apparent difference is identified between pre and post-vaccination carriage rates has also been reported in other studies [34,35,36].

### 4.1. Serotype Distribution

The top five serotypes identified in this study were serotypes 23B, 6B, 23F, 19F, and 13, of which serotype 23F, 19F, and 6B have been previously described as part of the top seven serotypes that cause IPD among children in Africa and globally [37]. Several studies have reported the emergence of serotype 23B in carriage after PCV vaccination [20,38]. Consequently, serotype 23B has been implicated in IPD [2,39]. Interestingly, serotypes 23B, 35B, and 38 detected in our study have already been reported by Renner et al. (2019) to be causal pathogens of IPD among children under five years in Ghana [7]. However, serotype 1 which is the number one cause of IPD in Ghana [40,41] was not detected among the strains of this study, consistent with other published carriage data from Ghana [10,20,42,43]. We did not detect serotype 19A, the eighth most prevalent serotype worldwide [37] but rather serotype 19B was detected.

The NVT coverage of 61.6% concurs with 65.4% obtained in a post PCV study among healthy children in Accra, Ghana [20]. This synonymous finding confirms that serotype replacement has occurred in the study population and could be a reflection of the larger Ghanaian population. These data further suggest that NVT could become candidates for IPD in Ghana in the near future. Serotype replacement has been identified in many countries where PCV vaccination has been introduced [4,26,32]. On the African continent, countries that have so far introduced PCV in their routine immunization programs have also reported increased NVT in different populations [44,45].

Despite the substantial increase in non-PCV13 serotypes, we also observed the persistence of PCV13 serotypes. The persistence of PCV13 vaccine serotypes (VT) after PCV vaccination has been reported widely [4,6,7,20,29]. Similar to our findings, the PCV13 coverage of 38.4% agrees with 34.6% obtained in an earlier study [20]. When this coverage rate is compared to the pre-vaccination coverage rate of 48% [20], there seems to be a decline in PCV13 coverage. Even though PCV vaccination has not had a significant impact on the overall carriage rate, it has impacted greatly on the serotype distribution. We observed an overall VT carriage of 11.3% which is comparable to 16.5% and 18.7% obtained by two separate studies in Malawi [6,44]. It is therefore not surprising to find VT causing 58.3% of IPD among children in Ghana [7]. However, the residual VT carriage observed in this study contrasts what has been reported from high-income countries [30,32,38]. The disparity in data on residual VT carriage exposes the lack of data on the long term effects of the widely used PCV13 schedule of 3 + 0 in many developing countries. Countries such as Canada, Denmark, and The Netherlands have registered drastic decline in overall VT carriage partly because they used different vaccination schedules that included a booster [30,35,38]. Recent evidence supports the advocacy for use of ‘reduced-dose PCV schedules plus a booster’ for countries where children receive 3 + 0 doses of PCV13 [4]. Additionally, PCV doses without booster have been shown to elicit a waning immunity and in some cases promoted IPD [46]. It is therefore not without reason that Australia recently switched from a 3 + 0 to 2 + 1 in order to reap the full benefits of a dosing schedule with a booster [46]. The involvement of serotype 1 (PCV13 serotype) in pneumococcal meningitis in Ghana undermines the achievement of herd immunity after PCV vaccination and supports the call for variation in PCV doses [40,41]. It is therefore important to evaluate the impact of booster doses over a wider age group on schedules with and without a booster.

### 4.2. Antibiotic Resistance

Susceptibility to ceftriaxone was 97.4% similar to other published data [40,47,48]. Comparing pre-vaccination ceftriaxone non-susceptibility of 0–5.2% [10,49] to post-vaccination rate of 2.6% from this study and 3.7% [20] confirms no apparent change in ceftriaxone non-susceptibility in the face of PCV vaccination. This means ceftriaxone can still be used in treating pneumococcal infections. A recent study portrayed the use of ceftriaxone as a first-line drug for treating community-acquired respiratory infections [50]. We, therefore, warn that if this worrying trend continues, Ghana could be faced with ceftriaxone non-susceptibility, not only to pneumococci but also to other bacteria. PCV vaccination appears not to have had any impact on the persistent resistance to cotrimoxazole, since the figures remain largely unchanged when compared to data from the pre-vaccination era [10,49]. The sustained resistance to cotrimoxazole observed in this study has been reported from countries where cotrimoxazole has been used extensively [48,51,52]. Possible factors that could be contributing to this trend are the ease of accessibility to cotrimoxazole as it is an over the counter antibiotic in Ghana [53] and its extensive use in treating respiratory infections. Cotrimoxazole also doubles as a prophylactic drug for people infected with HIV in Ghana and several African countries [51,54,55]. Additionally, Ghana, among other African countries, continues to administer sulfadoxine/pyrimethamine to pregnant women to prevent malaria-related adverse outcomes [54,56]. Both cotrimoxazole and sulfadoxine/pyrimethamine act as inhibitors to enzymes in the metabolic pathway of folic acid synthesis. The impact of the continuous use of sulfadoxine/pyrimethamine on cotrimoxazole-resistant pneumococci has been reported [55,57]. The tetracycline resistance of 57% observed in this study is consistent with previous reports (63% and 66.7%) from Ghana [20], and countries contributing data to the global antibiotic surveillance network [58]. Tetracycline, locally known as ‘red and yellow’ like other antibiotics in Ghana can be acquired easily without prescription even though this practice is illegal [53]. The ease of accessibility serves as the bedrock for self-medication resulting in abuse of such antibiotics [59]. These reasons coupled with strong antibiotic selective pressure and the possible acquisition of resistance genes from other bacteria in the respiratory tract fuel the flourishing resistance to tetracycline.

The small proportion of strains resistant to erythromycin is consistent with previous post PCV reports [20]. However, compared to pre-vaccination data [10,49], erythromycin resistance is still low and similar to reports from other African countries [51,60]. The low erythromycin resistance could be explained by the infrequent prescription among children in Ghana [42,61]. However, the recent introduction of mass distribution of azithromycin as a prophylactic measure towards yaws elimination [62] among a section of the Ghanaian population could enhance macrolide resistance among pneumococcal strains in the near future, an effect that is already evident in neighboring Burkina Faso [63].

In the present study, 35.8% of the isolates were resistant to penicillin suggesting an attributable effect from PCV13 vaccination. This rate is not different from what has been reported by other post-vaccination studies carried out among different populations in Ghana [20,42]. Meanwhile, post PCV studies from Tanzania, USA, and Brazil reports of penicillin resistance rates of 41%, 37.1%, and 44%, respectively [28,64,65]. Compared to pre-vaccination rates of 45% and 63%, PCV has achieved one of its expected results, i.e., to reduce the burden of penicillin resistance [66]. However, we must add that the detection of penicillin resistance among NVTs needs to be monitored in order to achieve the full benefit of PCV vaccination. Furthermore, we report a drastic decline in MDR strains when we compare the MDR rate of 28.5% from our study and 20.5% by Dayie et al. [20] with pre-vaccination rates of 72% and 87% [10,49]. This indicates that PCV13 vaccination has had an effect on the overall MDR among pneumococcal isolates from Ghana. Even though 80% of the MDR strains were NVT, we predict the possibility of a continual decline of MDR in the future as VTs decline with a corresponding reduction in penicillin resistance.

### 4.3. Virulence Genes

Protein-based vaccines have been proposed as an alternative to PCVs with the hope that they will mitigate the effect of serotype replacement and cover all pneumococci irrespective of serotypes. Indeed, some pneumococcal proteins have been identified as promising vaccine candidates [14,16,67]. Interestingly, this is the first study to determine the prevalence of virulence genes among carriage isolates from Ghana. The *psrP* gene, located on a genomic island, has previously been described to aid lung cell attachment, enhance bacterial aggregation, and promote the formation of biofilms. This and its immunogenicity renders *psrP* to a promising vaccine candidate [15]. In the present study, we found *psrP* among 62.9% of the isolates, which is similar to 51.7% in a Spanish study [17]. A slightly lower prevalence of 45.7% was found among unvaccinated children in Peru [68]. However, a much higher prevalence was detected among 88% of serotype 1 strains from France compared to 3% from selected African countries [69]. Similar to other studies, *psrP* was detected in nearly all serotypes reported in this study, and therefore, no association was found between *psrP* and VT (OR 0.743, 95% CI 0.38–1.46). In general, we did not observe a significant relationship between *psrP* and MDR (OR 0.25, 95% CI 0.12–0.53) because of the high prevalence of *psrP* among the isolates. This finding aligns with observations made in previous studies [17,70]. The ability of PcpA to elicit strong immune response has provided sufficient evidence for its inclusion in potential bivalent and trivalent vaccines [17]. The *pcpA* gene was widely distributed in serotypes found in this study resulting in a prevalence of 87%, which is consistent with 92.7% by Selva and colleagues [17]. In contrast, there was a huge disparity between *pcpA* prevalence among serotype 1 strains originating from Africa and those from Europe and other continents [16,69]. Based on the high *pcpA* prevalence determined in this study, we predict that vaccine formulations with *pcpA* in combination with other protein candidates will be beneficial to Ghanaian children.

In TIGR4, the pilus-1 has been shown to enhance adhesion, thereby promoting colonization, and prevent habitation by other competing microflora [18]. The pilus-1 was only detected in approximately 25% of pneumococcal isolates in various populations before the widespread introduction of PCV7, with no difference in prevalence among carriage and invasive disease [18,71]. *PI-1* was detected in serotypes 6B, 19F, 9V, 6A, 13, and 11A, and *PI-2* in serotypes 19F and 23B. However, the presence of both *PI-1* and *PI*-*2* was identified in only serotype 19F isolates. Because the majority of the piliated serotypes from this study were covered by PCV13, we expect that continuous use of the vaccine can eradicate these piliated VT from the population. The generally low prevalence of the pilus islets found in this study is in agreement with others [51,72]. Furthermore, the emergence of piliation in NVTs as seen in other studies [72,73] could be of concern to Ghana as they could expand and emerge in pneumococcal disease. Additionally, the pilus islets were significantly associated with MDR and PCV13 serotypes (*p* < 0.001) which is in agreement with previous studies [71,74]. A decline of pilus prevalence after PCV introduction has been observed among American Indian children [19,71]. However, because there are no data on pilus prevalence on isolates collected from Ghana before PCV13 introduction, we cannot attribute the low prevalence observed in this study entirely to the use of PCV13 in Ghana. Hence, the generally low prevalence of the pilus islet among the isolates underscores the need to re-examine its inclusion in protein-based vaccines, as it may offer very little additional protective benefit in African countries like Ghana.

## 5. Conclusions

This study provides evidence of the significant impact of PCV vaccination in Ghanaian children. However, there is an increase in NVT despite the persistence of PCV13 serotypes and therefore our results warrants continuous monitoring. Continuous use of PCVs could further accelerate the declining rate of penicillin-resistant and MDR strains. The inclusion of PcpA and PsrP in pneumococcal protein-based vaccines could be beneficial for Ghanaian children.

## Figures and Tables

**Figure 1 microorganisms-08-01987-f001:**
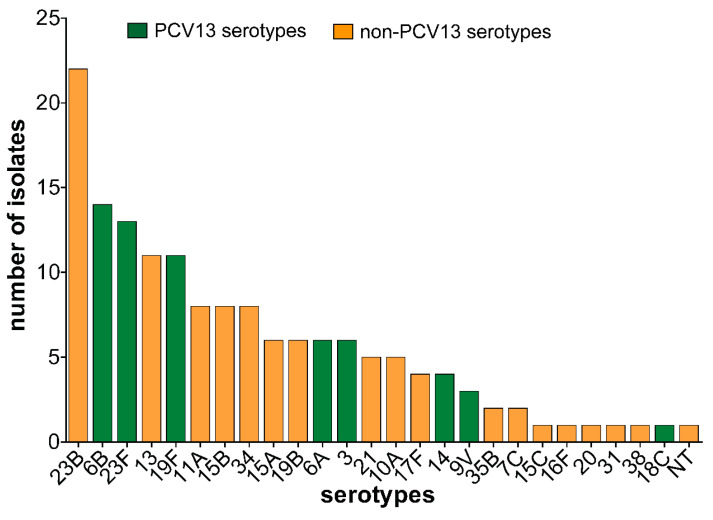
Distribution of vaccine and non-vaccine serotypes. Non-PCV13 serotypes are shown in orange color, while the PCV13 serotypes are shown in green color.

**Figure 2 microorganisms-08-01987-f002:**
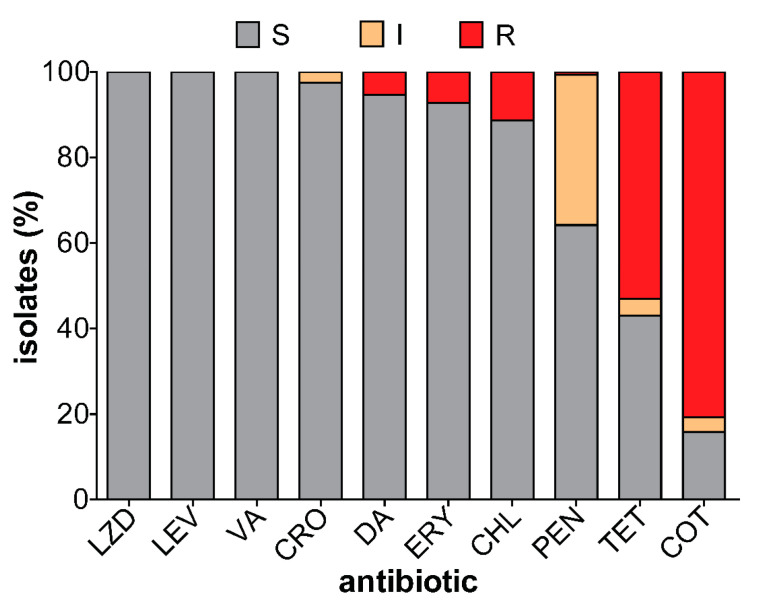
Antibiotic susceptibility patterns of pneumococcal isolates. Linezolid (LZD), Levofloxacin (LEV), Vancomycin (VA), Ceftriaxone (CRO), Clindamycin (DA), Erythromycin (ERY), Chloramphenicol (CHL), Penicillin (PEN), Tetracycline (TET), Cotrimoxazole (COT). S: susceptible (grey color), R: resistant (red color), I: intermediate resistance (orange color).

**Figure 3 microorganisms-08-01987-f003:**
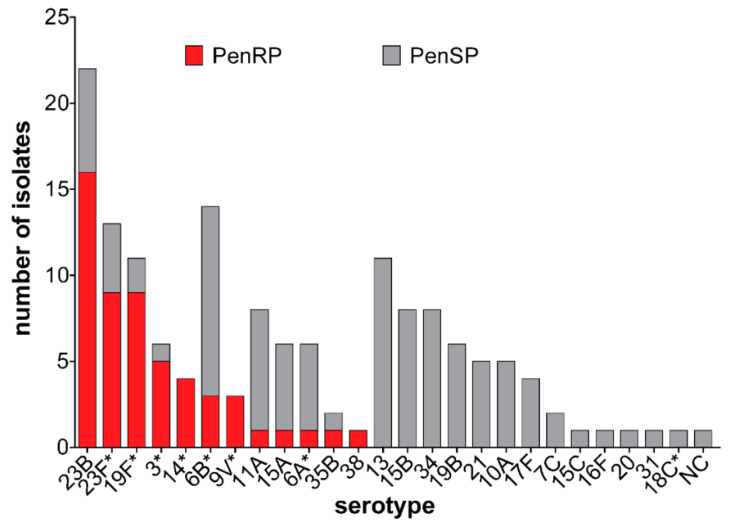
Penicillin susceptibility by serotype among pneumococcal strains. Serotypes were grouped into penicillin resistant pneumococci (PenRP) in red color and penicillin susceptible pneumococci (PenSP) in grey color. * PCV13 serotypes.

**Figure 4 microorganisms-08-01987-f004:**
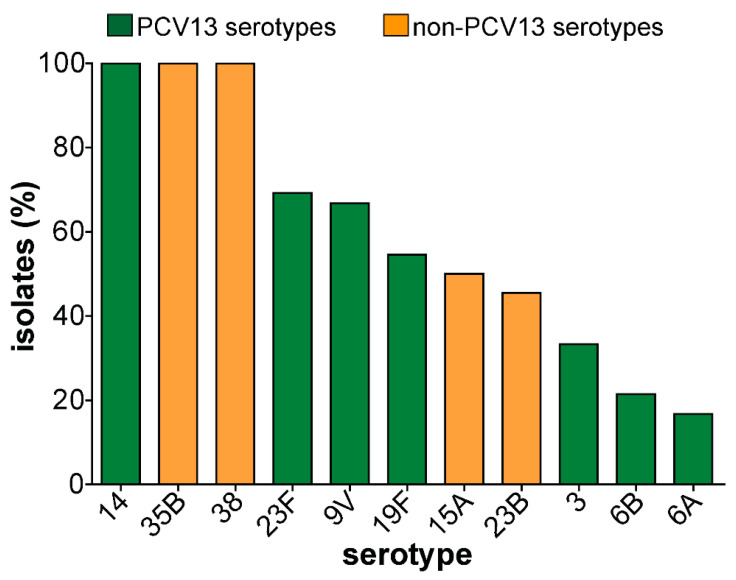
Distribution of multidrug resistance among isolates according to serotype. Pneumococcal isolates resistant to ≥3 different classes of antibiotics. Majority of the MDR isolates were PCV13 serotypes, except for serotypes 23B, 15A, 35B and 38. PCV13 serotypes are shown in green color and non-PCV13 serotypes are shown in orange.

**Figure 5 microorganisms-08-01987-f005:**
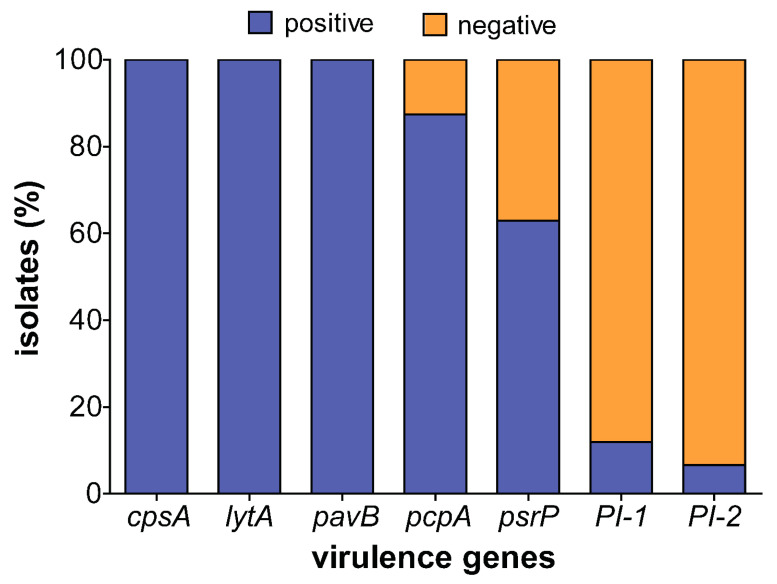
Prevalence of virulence genes among carriage isolates. More than half of the isolates tested positive for the *pcpA* and *psrP* genes while less than a quarter tested positive for the *PI-1* and *PI-2*. However, all the strains were positive for the core genome virulence genes including *pavB* and *lytA.*

**Table 1 microorganisms-08-01987-t001:** Participant characteristics and pneumococcal carriage.

Variable	Number of Children	Number of Carriers (%, 95% CI)	OR * (95% CI)(*p*-Value)
Total number of children	513	151 (29.4, 26–34)	
Gender			
Males	262	74 (28.2, 23–34)	1
Females	251	77 (30.7, 25–37	1.124 (0.77–1.64) (*p* = 0.546)
Source of participants			
Immunization clinic	100	27 (27, 18–36)	1
Kindergarten	413	124 (30, 26–34)	1.160 (0.71–1.89) (*p* = 0.552)
Age group (months)			
≤24	162	50 (30.9, 0.24–0.38)	1
>24	351	101 (28.8, 0.24–0.34)	0.905 (0.60–1.36) (*p* = 0.677)
1–6	6	2 (33, 21–88)	1
7–12	47	14 (30, 16–43)	1.29 (0.21–7.83) (*p* = 0.785)
13–18	54	18 (33, 20–46)	1.09 (0.45–2.63) (*p* = 0.846)
19–24	55	16 (29, 17–41)	1.29 (0.56–2.97) (*p* = 0.556)
25–30	98	22 (22, 14–31)	1.06 (0.45–2.46) (*p* = 0.902)
31–36	79	23 (29, 19–39)	0.74 (0.32–1.62) (*p* = 0.457)
37–42	26	8 (31, 12–50)	1.06 (0.48–2.32) (*p* = 0.892)
43–48	45	17 (38, 23–53)	1.14 (0.41–3.22) (*p* = 0.801)
49–54	53	17 (32, 19–45)	1.56 (0.66–3.70) (*p* = 0.312)
55–59	50	14 (28, 15–41)	1.21 (0.52–2.83) (*p* = 0.652)
Total	513	151 (29.4, 25–34)	

* OR: odds ratios, CI: confidence intervals, and *p*-values were calculated using logistic regression and two tailed Fisher’s Exact Test. A *p*-value < 0.05 was considered significant.

**Table 2 microorganisms-08-01987-t002:** Univariate and multivariate analysis of factors associated with multidrug resistance (≥3 antibiotics).

Variable	No./Total (%)	Crude OR * (95% CI)	*p*-Value	Adjusted OR * (95% CI *)	*p*-Value
Sex					
Male	17/74 (23)				
Female	26/77 (33.8)	1.709 (0.83–3.50)	0.153		
Age group					
≤24 months	15/50 (30)				
≥24 months	28/101 (72.3)	0.895 (0.43–1.89)	0.849		
Vaccine type					
non-PCV13	16/93 (17.2)				
PCV13	27/58 (46.6)	4.192 (1.99–8.84)	0.000	3.945 (1.68–9.26)	0.002
Virulence genes					
*pcpA* negative	6/19 (31.6)				
*pcpA* positive	37/132 (28.0)	0.844 (0.29–2.34)	0.788	1.036 (0.29–3.62)	0.956
*psrP* negative	26/56 (46.4)				
*psrP* positive	17/95 (17.9)	0.251 (0.12–0.53)	0.001	0.244 (0.11–0.55)	0.001
*PI-1* negative	35/133 (26.3)				
*PI-1* positive	8/18 (44.4)	2.240 (0.82–6.13)	0.161	1.168 (0.33–4.12)	0.809
*PI-2* negative	37/141 (26.2)				
*PI-2* positive	6/10 (60)	4.216 (1.13–15.78)	0.032	1.690 (0.38–7.48)	0.489

* OR: odds ratio, CI: confidence interval.

**Table 3 microorganisms-08-01987-t003:** Distribution of virulence genes among the different serotypes.

Serotype	*pcpA*	*psrP*	*PI-1*	*PI-2*	Total Number
23B	21	8	0	1	22
6B	7	11	7	0	14
23F	11	5	0	0	13
13	11	10	1	0	11
19F	9	6	5	9	11
11A	8	7	1	0	8
15B	8	6	0	0	8
34	6	7	0	0	8
15A	5	6	0	0	6
19B	6	5	0	0	6
3	6	2	0	0	6
6A	6	6	1	0	6
21	4	5	0	0	5
10A	5	1	0	0	5
14	4	2	0	0	4
17F	4	0	0	0	4
9V	3	1	3	0	3
35B	2	2	0	0	2
7C	0	2	0	0	2
15C	1	0	0	0	1
16F	1	1	0	0	1
18C	1	1	0	0	1
20	1	1	0	0	1
31	1	1	0	0	1
38	1	1	0	0	1
NT	0	0	0	0	1

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
