# Peer review of "Post-Vaccination Streptococcus pneumoniae Carriage and Virulence Gene Distribution among Children Less Than Five Years of Age, Cape Coast, Ghana"

_microorganisms, 2020, doi:10.3390/microorganisms8121987_

Round 1

Reviewer 1 Report

The manuscript by Mills et al. examines pneumococcal serotype, antibiotic resistance, and virulence gene distribution in Ghana after introduction of the conjugate pneumococcal vaccine. Over 500 samples from children under the age of five whom had been previously vaccinated were taken and tested for the presence of pneumococci. Close to a third of the samples contained resident pneumococci and were screened for serotype by quelling and multiplex PCR. Resistance was determined through standard clinical methods and specific gene targets were amplified to identify presence in the pneumococcal samples. As seen in other studies non-vaccine serotypes were prevalent in the isolates, but with a large portion of vaccine serotypes still present. Rates if drug resistance were similar to other studies conducted in the region and is attributed to local use of specific antibiotics. An interesting aspect of the current study is the examination of different virulence genes in the isolates obtained, which has been performed in other regions around the world but not Ghana.

Overall the manuscript is very well written and the techniques used were appropriate to adequately address the topic of the paper. As written there is very little that needs to be altered to increase the quality.

The only substantial issue with the current manuscript is the presentation of the data for multidrug resistance in figure 4. There were 10 and 9 multidrug resistant strains of serotype 23B and 23F respectively. As presented with just the total numbers it seems like 23B is more prevalent as a multidrug resistant serotype, but when total isolates are accounted for this is not the case. Of the 23B serotype 45% were multidrug resistant compared to 69% of 23F being multidrug resistant. While the data presented is accurate the way it is depicted makes 23B seem to have higher rates of multidrug resistance. Therefore, altering the figure to percentage of total isolates would be a more direct way to indicate resistance.

In relation to the previous concern is that a little over half of the isolates were resistant to 4 of the drugs tested, but is not discussed besides a brief mention in Ln 241. An extra sentence indicating if certain serotypes are more commonly resistant to 4 antibiotic as opposed to 3 or if the distribution is the same would be beneficial.

Author Response

Authors response to referee´s comments on microorganisms-1011799

Manuscript title:

Post-vaccination Streptococcus pneumoniae carriage and virulence gene distribution among children less than five years of age, Cape Coast, Ghana

Response:

We thank the reviewers for their constructive and insightful comments. We have considered these comments and addressed them in the main manuscript. The `track changes’ function in Microsoft word has been used to illustrate the modifications in the main manuscript submitted for the Reviewers. The final manuscript is submitted without indicating the changes.

Reviewer 1

The manuscript by Mills et al. examines pneumococcal serotype, antibiotic resistance, and virulence gene distribution in Ghana after introduction of the conjugate pneumococcal vaccine. Over 500 samples from children under the age of five whom had been previously vaccinated were taken and tested for the presence of pneumococci. Close to a third of the samples contained resident pneumococci and were screened for serotype by quelling and multiplex PCR. Resistance was determined through standard clinical methods and specific gene targets were amplified to identify presence in the pneumococcal samples. As seen in other studies non-vaccine serotypes were prevalent in the isolates, but with a large portion of vaccine serotypes still present. Rates if drug resistance were similar to other studies conducted in the region and is attributed to local use of specific antibiotics. An interesting aspect of the current study is the examination of different virulence genes in the isolates obtained, which has been performed in other regions around the world but not Ghana.

Overall the manuscript is very well written and the techniques used were appropriate to adequately address the topic of the paper. As written there is very little that needs to be altered to increase the quality.

The only substantial issue with the current manuscript is the presentation of the data for multidrug resistance in figure 4. There were 10 and 9 multidrug resistant strains of serotype 23B and 23F respectively. As presented with just the total numbers it seems like 23B is more prevalent as a multidrug resistant serotype, but when total isolates are accounted for this is not the case. Of the 23B serotype 45% were multidrug resistant compared to 69% of 23F being multidrug resistant. While the data presented is accurate the way it is depicted makes 23B seem to have higher rates of multidrug resistance. Therefore, altering the figure to percentage of total isolates would be a more direct way to indicate resistance.

Response:

We thank the reviewer for pointing out this observation. The proportion of MDR among each serotype has been expressed in percentages as suggested. Please see modified Figure 4.

In relation to the previous concern is that a little over half of the isolates were resistant to 4 of the drugs tested, but is not discussed besides a brief mention in Ln 241. An extra sentence indicating if certain serotypes are more commonly resistant to 4 antibiotic as opposed to 3 or if the distribution is the same would be beneficial.

Response:

Based on the reviewer’s suggestion, we have included a sentence (see line 279 in marked manuscript or line 243 in the final version) to further describe the serotype composition of the isolates within the ≥3 and ≥4 antibiotics.

Reviewer 2 Report

This manuscript describes, in general, the PCV13 post-vaccination carriage and distribution of virulence genes of Streptococcus pneumoniae in Ghana children. Overall, there is useful data, quite well presented. The manuscript requires only minor modifications.

In introduction section the following sentence is not clear: In these studies, circulating strains including serotypes……….were characterized by marked multidrug resistance (lines 64-65)

Methods: Table 1 should be included in the supplement.

Discussion section is too long (almost 4 pages) and should be markedly shortened as this is original paper not review. In  such a long discussion scientific soundness is lost.

Author Response

Authors response to referee´s comments on microorganisms-1011799

Manuscript title:

Post-vaccination Streptococcus pneumoniae carriage and virulence gene distribution among children less than five years of age, Cape Coast, Ghana

Response:

We thank the reviewers for their constructive and insightful comments. We have considered these comments and addressed them in the main manuscript. The `track changes’ function in Microsoft word has been used to illustrate the modifications in the main manuscript submitted for the Reviewers. The final manuscript is submitted without indicating the changes.

Reviewer 2

This manuscript describes, in general, the PCV13 post-vaccination carriage and distribution of virulence genes of Streptococcus pneumoniae in Ghana children. Overall, there is useful data, quite well presented. The manuscript requires only minor modifications.

In introduction section the following sentence is not clear: In these studies, circulating strains including serotypes……….were characterized by marked multidrug resistance (lines 64-65)

Response:

We have modified the sentence to ensure clarity; please see lines 64-65

Methods: Table 1 should be included in the supplement.

Response:

We have removed table 1 from the main manuscript. Table 1 contains the list of oligonucleotides used for the molecular typing of the virulence genes from the main manuscript and is shifted to the supplementary section. Please see section on supplementary materials.

Discussion section is too long (almost 4 pages) and should be markedly shortened as this is original paper not review. In such a long discussion scientific soundness is lost.

Response:

We have considered the comment of the reviewer and have therefore reduced the content of the discussion. All the changes can be tracked with the track changes function in MS word (PDF file labelled). We modified some of the sentences to make them shorter and removed, in addition, non-essential information. Pease see discussion lines 278-426.